# A Silent Threat in Post-Tuberculosis Patients: Chronic Pulmonary Aspergillosis Survey in Multiple Regions of Indonesia (I-CHROME Study)

**DOI:** 10.3390/jof11050329

**Published:** 2025-04-22

**Authors:** Anna Rozaliyani, Findra Setianingrum, Fathiyah Isbaniah, Heidy Agustin, Raden Rara Diah Handayani, Rosamarlina Syahrir, Siti Pratiekauri, Robiatul Adawiyah, Hesti Setiastuti, Mohammad Nizam Erhamza, Retno Ariza S. Soemarwoto, Irvan Medison, Deddy Herman, Avissena Dutha Pratama, Jatu Apridasari, Jani Jane, Soedarsono Soedarsono, Tutik Kusmiati, Mufidatun Hasanah, Diah Adhyaksanti, Winda Sofvina, Ammar A. Hasyim, Chris Kosmidis, David W. Denning

**Affiliations:** 1Department of Parasitology, Faculty of Medicine, Universitas Indonesia, Jakarta 10430, Indonesia; drfindra@gmail.com (F.S.); robiatul.adawiyah01@ui.ac.id (R.A.); ammarhasyim26@gmail.com (A.A.H.); 2Pulmonary Mycosis Centre, Jakarta 10430, Indonesia; windasofvina2706@gmail.com; 3Grha Permata Ibu Hospital, Depok 16425, Indonesia; nizpulmo@gmail.com; 4The Indonesian Society of Respirology (Perhimpunan Dokter Paru Indonesia), Jakarta 13340, Indonesia; fathiyah21@gmail.com (F.I.); heidy_agst@yahoo.com (H.A.); 5Department of Pulmonology and Respiratory Medicine, Persahabatan Central General Hospital, Universitas Indonesia, Jakarta 13230, Indonesia; 6Department of Pulmonology and Respiratory Medicine, Indonesia University Hospital, Depok 16425, Indonesia; diahzulfitri@yahoo.com; 7Department of Pulmonology and Respiratory Medicine, Sulianti Saroso Infectious Disease Hospital, Jakarta 14340, Indonesia; rosa_pulmo@yahoo.co.id (R.S.); ssoedarsono@gmail.com (S.S.); 8Sulianti Saroso Infectious Disease Hospital, Jakarta 14340, Indonesia; sitipratiekauri@yahoo.com; 9The Indonesian Society for Medical Specialist in Clinical Parasitology (Perhimpunan Dokter Spesialis Parasitologi Klinik Indonesia, PDSPARKI), Jakarta 10430, Indonesia; 10Tangerang City General Hospital, Tangerang 15111, Indonesia; hestissdr@gmail.com; 11Abdul Moeloek General Hospital, Bandar Lampung 35112, Indonesia; arizapulmo@gmail.com; 12Department of Pulmonology and Respiratory Medicine, RSUP Dr. M. Djamil, Padang 25129, Indonesia; irvanmds67@gmail.com; 13Madina General Hospital, Bukittinggi 26117, Indonesia; deddykaterine@gmail.com; 14Kariadi General Hospital, Semarang 50244, Indonesia; avissena_dp@yahoo.com; 15Moewardi General Hospital, Solo 57126, Indonesia; jatuparu@gmail.com; 16Syaiful Anwar General Hospital, Malang 65112, Indonesia; yani.sugiri@yahoo.com; 17Dr. Soetomo General Hospital, Surabaya 60286, Indonesia; tutik.kusmiati93@gmail.com; 18Kanujoso General Hospital, Balikpapan 76115, Indonesia; mufida09hasanah@gmail.com; 19Beriman General Hospital, Balikpapan 76113, Indonesia; d.adhyaksa.dr@gmail.com; 20Manchester Academic Health Science Centre, Division of Infection, Immunity and Respiratory Medicine, School of Biological Sciences, Faculty of Biology, Medicine and Health, University of Manchester, Manchester M23 9LT, UK; chris.kosmidis@manchester.ac.uk (C.K.); ddenning@manchester.ac.uk (D.W.D.)

**Keywords:** chronic pulmonary aspergillosis, post-tuberculosis, IgG antibodies

## Abstract

Background: A significant complication among post-tuberculosis patients is chronic pulmonary aspergillosis (CPA), with prevalence and outcomes varying by region. This study aimed to explore the epidemiology, clinical characteristics, and microbiological profiles of 219 post-tuberculosis patients with persistent respiratory symptoms and lung cavities in Indonesia. Methods: The patients were divided into CPA (*n* = 144) and non-CPA (*n* = 75) groups. This cross-sectional study diagnosed CPA in post-tuberculosis patients using ERS/ESCMID criteria, integrating clinical, radiological, and fungal assessments. Serological tests for *Aspergillus*-specific IgG were conducted using immunochromatographic (ICT) and ELISA on serum samples. Sputum specimens were used in parallel for fungal culture, and radiological evaluations (e.g., chest X-rays or CT scans) were performed to identify typical CPA features such as cavitation and fibrosis. Results: Persistent cough was significantly more common in CPA patients (83.3%, *p* = 0.015), highlighting its role as a clinical indicator for CPA. Radiological infiltrates were found in 165 patients (75.3%); critical diagnostic markers of CPA were cavitation and pericavitary fibrosis. *Aspergillus*-specific IgG testing demonstrated high diagnostic utility, with positivity rates of 69.4% for ICT and 63.2% for ELISA among CPA patients. Among those with infiltrates, a positive *Aspergillus* culture was not more common (*p* > 0.05), whereas *Aspergillus* IgG was more often raised (*p* = 0.037), as was a positive ICT (*p* = 0.021). Regional analysis revealed a higher CPA burden in Region 1 (75%) compared to Region 2 (56%, *p* = 0.003), with *Aspergillus fumigatus* and *Aspergillus niger* predominating in Region 1. Conclusions: These findings highlight the importance of comprehensive approaches and region-specific CPA management strategies in Indonesia.

## 1. Introduction

Chronic pulmonary aspergillosis (CPA) represents a critical global health challenge, particularly in regions where pulmonary tuberculosis (TB) is highly prevalent [1,2,3]. CPA is a chronic, progressive lung disease caused by infection with *Aspergillus* species, primarily *Aspergillus fumigatus*, and it often arises as a complication in patients with pre-existing lung conditions, most notably those recovering from TB [1,4,5]. In many cases, CPA develops in individuals who have residual lung cavities, damaged lung tissue, or chronic obstructive pulmonary disease (COPD) following TB infection, making this population particularly vulnerable [4,6]. This interaction between CPA and TB sequelae presents a considerable public health burden in low- and middle-income countries, where TB prevalence remains high [7,8].

Indonesia has one of the highest TB burdens in the world and faces growing concerns over the incidence and management of CPA among its post-TB population [6]. The country’s tropical climate, combined with widespread exposure to environmental *Aspergillus* spores, creates a conducive environment for the development of CPA [9]. Nevertheless, CPA is frequently underdiagnosed and commonly mistaken for TB recurrence or other chronic respiratory diseases because of overlapping symptoms such as chronic cough, hemoptysis, and fatigue [10]. This diagnostic uncertainty of CPA contributes to poor outcomes and significant mortality in affected patients by delaying effective treatment and increasing the risk of disease progression.

Nonspecific symptoms of CPA, which are shared by a variety of pulmonary conditions, present a clinical challenge in diagnosing CPA [11]. For instance, a characteristic of CPA is a persistent cough, which is also frequently observed in tuberculosis and other respiratory infections, complicating the differential diagnosis [12]. Additionally, traditional diagnostic methods, such as imaging and microbiological cultures, can be inconclusive or unavailable in resource-limited settings [13]. Radiological findings, such as cavitation, pericavitary fibrosis, and fungal balls (aspergillomas), are particularly important for CPA diagnosis, but access to imaging tools varies significantly between urban and rural regions [14]. Therefore, enhancing diagnostic accuracy is crucial, which can be achieved by identifying the reliable diagnostic biomarkers, such as *Aspergillus*-specific IgG antibodies [15]. The development and validation of serological tests, including immunochromatographic tests (ICT) and enzyme-linked immunosorbent assays (ELISA), offer a promising solution for resource-constrained settings [16,17,18].

An additional challenge in managing CPA is the regional variability in its prevalence and outcomes [19]. Factors such as varying environmental exposure to *Aspergillus* spores, differences in healthcare infrastructure, and socio-economic disparities contribute to notable variations in CPA incidence and mortality in Indonesia [20]. Densely populated urbanized regions such as Jakarta and West Java may experience higher CPA prevalence due to greater environmental exposure and better diagnostic facilities. In contrast, rural areas may have lower reported cases, which could reflect underdiagnosis rather than a genuinely lower burden [21]. Furthermore, disease presentation and management strategies may be influenced by regional variability in microbiological profiles, such as the predominance of *A. fumigatus* in urban areas and *A. niger* in rural settings [22,23,24].

Limited awareness among healthcare providers and inconsistent diagnostic protocols further exacerbate the underdiagnosis of CPA in Indonesia [3,25]. Many patients are diagnosed with CPA at an advanced stage of the disease, when symptoms have significantly worsened and irreversible structural lung damage has occurred. The lack of comprehensive diagnostic guidelines that incorporate clinical, radiological, and microbiological parameters is often the cause of this delay. While serological tests, particularly Aspergillus-specific IgG antibody detection, have shown high diagnostic utility, their use is not yet standardized nationwide. Expanding access to affordable and reliable diagnostic tools like ICT and ELISA could significantly improve CPA detection rates, particularly in remote regions.

This research is the first comprehensive investigation of CPA across multiple provinces in Indonesia, including both urban and rural regions. The study aims to identify key epidemiological, clinical, and microbiological characteristics of CPA by analyzing post-TB patients in this high-burden setting. In addition to microbiological evidence and serological testing, this study also investigates the use of radiological markers such as cavitation and fibrosis in diagnosing CPA. To provide a more in-depth picture of the CPA burden in Indonesia among post-tuberculosis patients, the study examines regional disparities in CPA proportion, microbiological profiles, and diagnostic characteristics across various provinces. This study reinforces the importance of early and accurate diagnosis in managing CPA. The findings highlight the urgent need to improve CPA detection and treatment, especially in Indonesia and other countries with a high burden of tuberculosis.

## 2. Materials and Methods

### 2.1. Study Design and Patient Selection

The study was conducted between June 2023 and March 2024, during which consecutive patients presenting with respiratory symptoms with a prior TB history to the chest clinics of 14 hospitals were enrolled. The inclusion criteria required patients to have respiratory symptoms such as cough, hemoptysis, shortness of breath, chest pain, or fatigue for three months. Patients meeting the exclusion criteria were excluded from the study. Exclusion criteria included (1) refusal to provide informed consent for participation or serological testing, (2) incomplete clinical or diagnostic data, (3) current diagnosis or ongoing treatment for active tuberculosis at the time of screening, (4) death prior to or during diagnostic evaluation, (5) pregnancy, (6) known HIV infection, (7) severe immunocompromised state (e.g., recent chemotherapy with complications or critical illness), and (8) use of systemic antifungal agents within the last one month. The study received ethical approval from the Ethics Committee of the Faculty of Medicine, Universitas Indonesia (KET-405/UN2.F1/ETIK/PPM.00.02/2023). Written informed consent was obtained from all participants before enrollment, per institutional protocol. Upon enrollment, a trained professional recorded the case details in a standardized case report form. Relevant investigations, including blood tests, chest imaging, and sputum examination, were conducted based on the discretion of the treating physicians. Serum samples were obtained from all participants and were evaluated using the LDBio lateral flow assay (LFA; LDBIO Diagnostics (4 avenue Joannes Masset-69009 Lyon, France; #Lot No. ASPG-015) and Asp-IgG assay manual ELISA (Bordier Affinity Products SA, Batiment Biokema, Chatanerie 2, 1023 Crissier, Switzerland; #Lot No. 2417A-R). Sputum cultures were conducted using high-volume methods.

### 2.2. Diagnosis of Chronic Pulmonary Aspergillosis

The diagnosis of CPA was established using the criteria set by the European Respiratory Society/European Society of Clinical Microbiology and Infectious Diseases (ERS/ESCMID) [26]. The diagnosis was based on clinical, radiological, and microbiological parameters (Table 1). Patients were required to have at least one of the following symptoms for more than three months: hemoptysis, cough, exhaustion, chest discomfort, or dyspnea. Radiographic findings consistent with CPA, such as cavitation and/or the presence of a fungal ball, were necessary for diagnosis. These criteria were adapted to suit the resource-constrained settings in which the study was conducted. Microbiological evidence included a positive serological result with *Aspergillus*-specific IgG levels, with a cut-off value of ≥0.821 milligrams of antibodies/liter considered positive [27]. Moreover, the identification of hyaline, septate hyphae on direct microscopy or the isolation of *Aspergillus* spp. in culture from respiratory samples further supported the diagnosis.

### 2.3. Serological Testing

The *Aspergillus* ICT IgG/IgM lateral flow assay (LFA) was used to test serum samples, following the manufacturer’s instructions. Cassettes were removed from storage at 4 °C, brought to room temperature, and labeled. A 15 µL serum sample was added to the sample well of each cassette using a calibrated micropipette with sterile disposable tips. Four drops of eluent were then added directly from the dropper. The cassettes were allowed to stand for 20 min, and the test results were read between 20 and 30 min after adding the eluent to the last cassette.

A positive result was characterized as having a well-defined black line at both the “Test (T)” and “Control INCL”de” markers, while a “weakly positive” result was classified as having a thin, diffuse grey line at the t“e”“T” marker. The turnaround time was between 30 to 45 min, which included pre-test centrifugation of blood samples, documentation, and processing. An alternative analysis was conducted using the Bordier Asp-IgG ELISA assay with a cut-off of ≥0.821, validated in the previous study [27].

### 2.4. Statistical Analysis

The demographic information gathered consisted of age, gender, body mass index (BMI), duration of signs and symptoms, and any underlying diseases. Fisher’s exact test or the Chi-squared test was applied to categorical variables. Continuous variables were compared using Student’s *t*-test. For all analyses in this study, a *p*-value of less than 0.05 was considered statistically significant. The statistical analysis was performed using IBM SPSS version 22 software.

## 3. Results

### 3.1. Classification of Study Subjects

A total of 221 patients, aged 18 years or older, with a prior history of TB, patients with negative TB, and respiratory symptoms lasting at least three months were screened for inclusion in the study. Of these, two patients were excluded for various reasons, including incomplete data and death. After applying these exclusion criteria, 219 patients were deemed eligible and enrolled for further analysis (Figure 1).

The 219 eligible patients underwent detailed diagnostic assessment, including evaluations of a mycological culture or serological testing for *Aspergillus*-specific IgG. The patients were divided into two main groups based on the outcomes of these thorough evaluations. The first group comprised 144 patients diagnosed with CPA according to established diagnostic criteria. The second group consisted of 75 patients who did not meet the diagnostic criteria for CPA and were categorized as non-CPA (Figure 1). This robust diagnostic approach ensured the precise identification of CPA cases and facilitated the implementation of targeted treatment strategies for the affected patients.

### 3.2. Demographics and Clinical Characteristics

The study enrolled 219 patients, of whom 144 (65.8%) were diagnosed with CPA and 75 (34.2%) were without CPA (non-CPA) (Table 2). The CPA group had a male predominance (56.9%), while the non-CPA group had an even higher proportion of males (66.7%) (*p* = 0.163).

The analysis of symptoms revealed that cough was significantly more common in CPA patients, with 92.4% reporting this symptom compared to 81.3% in the non-CPA group (*p* = 0.015; Table 2). This finding suggests that a persistent cough could be a potential indicator of CPA in patients with post-tuberculosis lung disease. The other symptoms—fatigue, dyspnea, chest pain, and hemoptysis—were similarly prevalent in both groups, as indicated by their respective *p*-values (*p* = 0.663, 0.619, 0.228, and 0.325). The median body mass index (BMI) was slightly lower in the CPA group (18.4; range 11.7–28.9) compared to the non-CPA group (19.7; range 10.4–31.6). However, this difference was not statistically significant (*p* = 0.135). The most common category of BMI in the CPA group was underweight, with 73 patients (50.7%).

Regarding underlying diseases, 109 (49.8%) had at least one underlying condition, including chronic obstructive pulmonary disease (COPD), hypertension, asthma, or diabetes mellitus. The remaining 110 (50.2%) patients had no documented underlying comorbidities. COPD was observed in 20.8% of CPA patients compared to 18.7% of non-CPA patients (*p* = 0.704). Similarly, hypertension, asthma, and diabetes mellitus showed no significant differences between the two groups, with p-values of 0.270, 0.142, and 0.803, respectively (Table 2).

### 3.3. Diagnostic Findings and Multimodal Approaches

A comprehensive integration of radiological, culture, and serological findings in the evaluation of CPA highlighted the complexity of diagnosis and the need for multimodal approaches (Table 3; Figure 2). The data showed concordant and discordant results between radiological findings and positive culture or serology, underscoring the limitations of relying on any diagnostic modality.

Among radiological findings, 165 patients (75.3%) had infiltrates. Of these, 39 patients (23.6%) showed positive cultures for *A. fumigatus*, while 62 patients (37.6%) and 68 patients (41.2%) were positive for *Aspergillus*-specific IgG and ICT, respectively. Significant associations were found between infiltrates and IgG (*p* = 0.037) or ICT (*p* = 0.021), indicating that serological tests can detect cases not confirmed by culture. This emphasizes the role of serology in diagnosing CPA when culture results are negative despite evident radiological abnormalities.

Cavitations were the most frequent radiological finding, seen in 191 patients (87.2%). Positive culture results for *A. fumigatus* were noted in 49 cases (25.6%), while serological positivity rates were 41.4% for IgG and 45.5% for ICT. Despite the high frequency of cavitations, no significant association was observed between radiological findings and *A. fumigatus* cultures (*p* = 0.371). However, serology remained valuable in identifying CPA cases even when the culture was negative, highlighting its diagnostic importance.

Pericavitary fibrosis was observed in 173 patients (79.0%), and 45 positive *A. fumigatus* cultures were demonstrated (26.0%); serological assays showed positivity in 76 cases (43.9%) for IgG and 79 cases (45.7%) for ICT, though without significant associations. Pleural thickening was identified in 95 patients (43.4%), with positive *A. fumigatus* cultures in 26 cases (27.4%), while serological positivity was comparable to other findings (43.3% for IgG and 42.1% for ICT). These findings reinforce the limited reliability of radiological features for CPA diagnosis, especially when fungal cultures are negative.

Nodules were noted in 30 patients (13.7%), with *A. fumigatus* cultured in seven cases (23.3%) and IgG and ICT positivity in 46.7% and 43.3%, respectively. Aspergillomas, observed in 24 patients (11.0%), were significantly associated with serology, showing IgG positivity in 66.7% (*p* = 0.008) and ICT positivity in 66.7% (*p* = 0.029). These findings highlight the diagnostic strength of serology in specific radiological patterns where culture sensitivity is low.

In 133 patients (60.7%), bronchiectasis was significantly associated with *A. flavus* cultures (*p* = 0.046), suggesting a unique relationship between structural lung abnormalities and colonization by non-*fumigatus Aspergillus* species. This finding indicates the need to further explore bronchiectasis as a potential risk factor or manifestation of CPA.

The CPA groups in this paper consisted of different groups varied by their culture and serology tests. The titer of *Aspergillus* Ig-G measured by ELISA showed the lowest value in the non-CPA group and, interestingly, in the CPA group with positive culture and negative serology. The highest *Aspergillus* IgG titers were reported from CPA sera with positive sputum cultures (Figure 3).

### 3.4. Geographic Distribution and Regional Variability

The geographic and regional analysis of CPA in Indonesia in this study reveals significant disparities in both the distribution and impact of the disease across various regions (Table 4). A notable variation can be seen in the mapping of CPA cases, with the highest proportion of CPA observed in Banten (85.4%), followed by DKI Jakarta (71.4%) and East Java (70%). Contributing factors to the relatively high proportion of CPA in these populous regions are increased environmental exposure, greater healthcare access, and more robust diagnostic capabilities. In contrast, regions such as West Sumatra and Central Java, each reporting 14 CPA cases, and East Kalimantan with seven cases, demonstrate a lower but still notable presence of the disease. Lampung, with only two reported cases, stands out as the region with the fewest CPA cases, potentially indicating a genuinely lower proportion or challenges in diagnosis and reporting.

When analyzing the regional distribution in a broader context, the study divided the patients into two major regions: Region I, which included Jakarta, Tangerang, and West Java, and Region II, encompassing Middle and East Java, Sumatra, and Kalimantan. The division into Region I (Jakarta, Tangerang, and West Java) and Region II (Central Java, East Java, Sumatra, and Kalimantan) was based on geographic proximity, socioeconomic characteristics, and differences in healthcare accessibility. Region I includes more urbanized and densely populated areas with advanced healthcare infrastructure and higher diagnostic capacity. Region II, in contrast, consists of more rural or less densely populated areas with relatively limited access to specialized diagnostic tools (Figure 4).

Of the 219 patients included in the study, 115 were from Region I, and 104 were from Region II. A statistically significant difference was found in the proportion of CPA between these regions. In Region I, 75% of the patients (86 out of 115) were diagnosed with CPA, whereas Region II had a lower proportion, with 56% (58 out of 104) of patients diagnosed with CPA (*p* = 0.003).

### 3.5. Aspergillus Species Across Region

*Aspergillus fumigatus* was the most frequently isolated species, which was found in 24.8% (54/219) of the total patient population (Table 5). *A. fumigatus* was more prevalent in Region I (33%, 38/115) than in Region II (15.5%, 16/104), with the difference reaching statistical significance (*p* = 0.003). *Aspergillus flavus* was identified in 9.6% (21/219) of patients, with no significant difference between Region I (12.2%, 14/115) and Region II (6.8%, 7/104) (*p* = 0.179).

*Aspergillus niger* was isolated in 20.2% (44/219) of patients, with a higher proportion in Region I (30.4%, 35/115) compared to Region II (8.7%, 9/104), and this difference was statistically significant (*p* < 0.001; Table 5). This result highlights the need for region-specific strategies in managing CPA, including targeted antifungal therapy and environmental interventions to reduce fungal exposure.

## 4. Discussion

The present study provides significant insights into the epidemiology, clinical characteristics, diagnostic challenges, and regional disparities of CPA in post-tuberculosis patients in Indonesia. As the first multicenter study involving multiple provinces in Indonesia, our findings underscore the importance of region-specific strategies for diagnosing and managing CPA, particularly in populations with high post-tuberculosis sequelae [1,21,29].

Our findings indicate that the gender distribution does not significantly differ between CPA and non-CPA patients, suggesting that gender alone is not a distinguishing factor in CPA prevalence within this population. This aligns with previous studies that attribute CPA risk to environmental exposures and underlying health conditions rather than gender [3,29,30]. Distinguishing CPA from pulmonary tuberculosis on clinical grounds alone is difficult or impossible. We did find that a persistent cough was more frequent in CPA patients compared to non-CPA patients, underscoring its potential role as a clinical marker for CPA in post-tuberculosis patients [21,31,32]. Other symptoms, such as fatigue, dyspnea, chest pain, and hemoptysis, were common in both CPA and non-CPA groups, showing no significant differences. These non-specific manifestations are typical of various pulmonary conditions, often leading clinicians to suspect bacterial etiologies initially. This diagnostic ambiguity underscores the necessity of incorporating specific serological and mycological tests into the diagnostic workup to improve early detection of CPA and avoid misdiagnosis [33,34].

Interestingly, the study found no significant difference in the proportion of underlying diseases such as COPD, hypertension, asthma, and diabetes mellitus between CPA and non-CPA patients. This contrasts with some prior studies that have suggested a higher prevalence of underlying pulmonary or systemic conditions in CPA patients [12,29,35]. The lack of significant differences in our study may reflect the relatively uniform health status of the post-tuberculosis population or potential differences in disease pathogenesis in this specific demographic.

Five out of 15 (33%) deceased CPA patients showed negative fungal culture but positive ICT and/or ELISA antibody tests, with the highest *Aspergillus* IgG titer. These findings highlight the utility of *Aspergillus* IgG antibody detection as a critical diagnostic tool for CPA, especially in resource-limited settings where advanced imaging and invasive diagnostic procedures may not be readily available [36,37]. If CPA were diagnosed only with the combination of clinical, radiology, and sputum culture, 70 (61.4%) cases would have been undetectable as CPA cases. These findings showed proof that serology tests such as ICT and/or ELISA could serve as an effective diagnostic method in endemic regions, especially ICT for rural areas with limited laboratory infrastructure. However, there were 29 CPA cases with negative serology for both ELISA and ICT methods. The sputum culture result from this group was dominated by *A. niger* (20 of 29 cases), but mostly mixed colonies with *A. fumigatus* (17 of 20 cases) and *A. flavus* (10 of 29 cases). The fungal species grown from the sputum culture might affect the variation of *Aspergillus* IgG titer, as most of the commercial products were used for the detection of IgG-specific *A*. *fumigatus.* The mean *Aspergillus* IgG titer from CPA cases positive for *A. fumigatus* was 2.8 mg/L, in contrast with other species (*A. flavus*, 1.4 mg/L and *A. niger*, 2.1). The serology test with specific antibody detection against *A. niger* might be required to assist the CPA cases caused by *A. niger* after we ruled out colonization. Also possible is that the attribution of CPA to *A, niger* is incorrect and it reflects either colonization or *Aspergillus* bronchitis [38], the latter especially common in bronchiectasis, another common sequela of TB. *A. flavus* has been a relatively uncommon cause of CPA but may be more common in SE Asia [39,40].

The study showed significant regional disparities in CPA proportion, with a higher burden observed in Region I (Jakarta, Tangerang, and West Java) compared to Region II (Middle and East Java, Sumatra, and Kalimantan). The significantly higher proportion of CPA in Region I may be attributed to various factors, including greater environmental exposure to *Aspergillus* species, urbanization, and differences in healthcare infrastructure that facilitate better case detection. The geographic variation in CPA proportion underscores the need for region-specific public health interventions, including improved environmental controls and targeted screening programs in high-prevalence regions [37]. In contrast to another study with a low prevalence of CPA [41], the relatively high proportion of CPA in our study is because the inclusion criteria already included clinical and radiological features of CPA.

Moreover, radiological findings in CPA patients offer essential insights into the disease’s diagnostic features and regional variations. These findings emphasize the role of radiological evaluation in diagnosing CPA and shaping management strategies. Key radiological criteria, such as cavitation and pericavitary fibrosis, are critical markers for CPA [42]. Regional differences, including the higher prevalence of infiltrates and bronchiectasis in different regions, highlight the necessity for diagnostic approaches considering geographic and environmental factors [43]. Combining radiological features with clinical and microbiological parameters improves diagnostic accuracy and supports more effective CPA management, especially in resource-constrained and diverse healthcare settings [44].

Cases that showed negative radiological features but had positive culture or serology results were noteworthy. For instance, among patients without cavitations, *A. fumigatus* cultures were positive in 17.9%, and serological positivity for IgG and ICT was 42.9% and 46.4%, respectively. This discordance highlights the limitations of relying solely on radiological features and emphasizes the importance of routine mycological and serological evaluations to prevent missed diagnoses.

Our study has several limitations. First, the type and number of hospitals and local recruitment teams’ capabilities in every province differ. This resulted in the variability of the case detection rate in different provinces. Technical problems regarding research permissions were very complicated in specific hospitals, which shortened the duration of patient recruitment. The CPA case numbers reported in our study are preliminary results and might not reflect the real burden of CPA in each province. Second, the sputum and blood from local hospitals were sent to the mycology laboratory of the Faculty of Medicine, Universitas Indonesia in Jakarta. This process sometimes took at least 24 h for the samples to reach our laboratory, especially those from different islands. The delayed sample processing might affect the results of some of the tests. However, precautions for this problem were taken, as we delivered mycology laboratory training for our local research team with procedures on how to package the sample with safety procedures.

## 5. Conclusions

In summary, this study highlights the critical role of comprehensive diagnostic approaches, including *Aspergillus* IgG antibody tests, in detecting CPA in post-tuberculosis patients. Regional differences in CPA proportion and outcomes highlight the need for public health interventions, including improved diagnostic access, environmental controls, and antifungal therapy in high-burden areas. Further research is needed to explore the environmental and population-specific factors driving these disparities and to develop effective strategies to reduce the burden of CPA across Indonesia.

## Figures and Tables

**Figure 1 jof-11-00329-f001:**
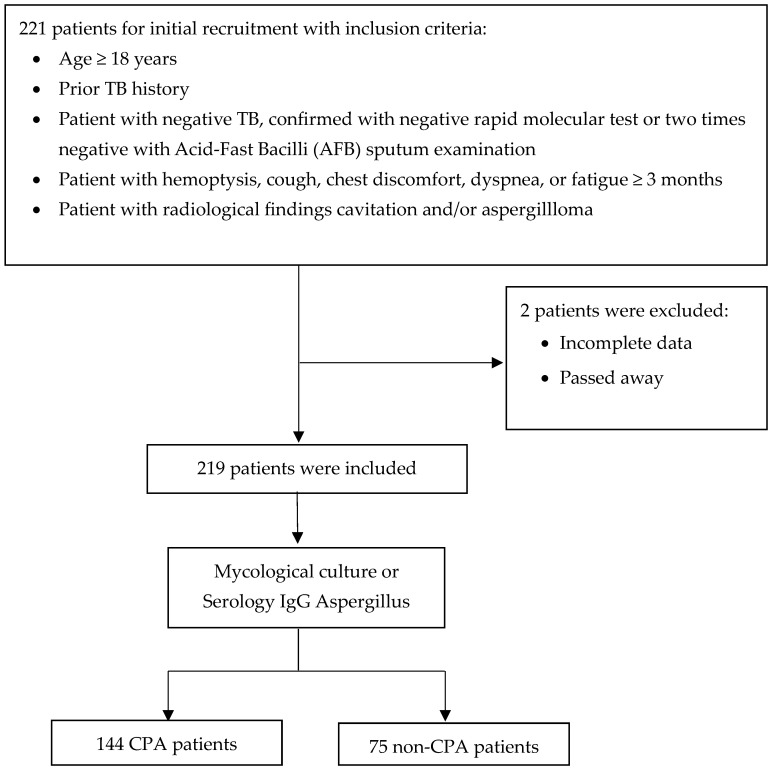
Flow diagram of patient recruitment, exclusions, and classification.

**Figure 2 jof-11-00329-f002:**
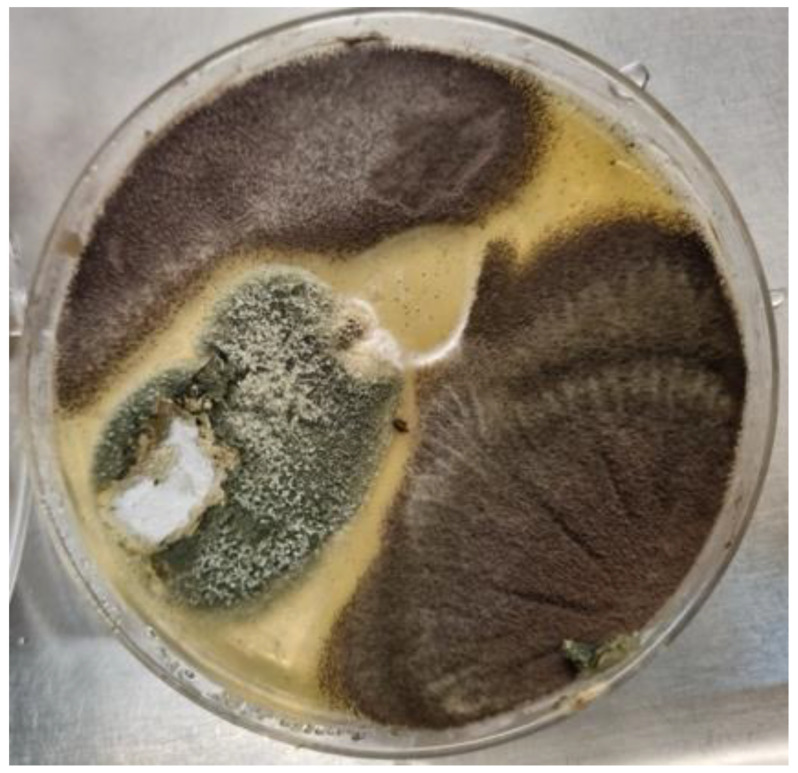
A culture plate of Sabouraud dextrose agar from a CPA patient. The dark brown colonies were consistent with *Aspergillus niger* complex. The grey-green colony shows white areas representing poorly sporulating sections within the colony (characteristic of many CPA isolates), and microscopic examination revealed *Aspergillus fumigatus* complex.

**Figure 3 jof-11-00329-f003:**
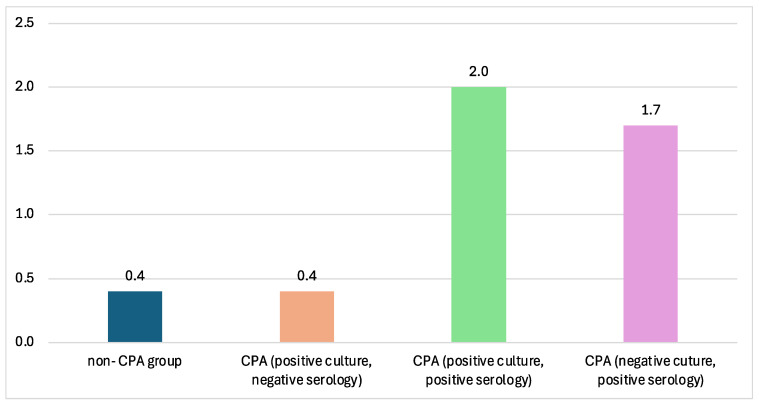
The mean ELISA titer across different CPA groups. The highest ELISA titer is from the CPA group with positive *Aspergillus* culture from sputum, with 2.0 mg/L.

**Figure 4 jof-11-00329-f004:**
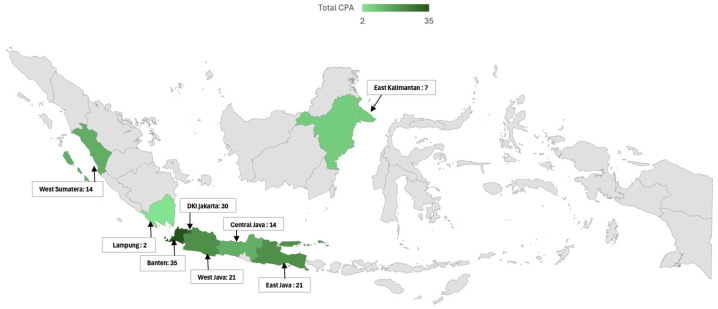
Geographic distribution of chronic pulmonary aspergillosis (CPA) cases in Indonesia. This map illustrates the geographic distribution of CPA cases across various regions of Indonesia, highlighting regional differences in the proportion of CPA. The total number of CPA cases is indicated for each region. The indicated map color is green in the study area, and grey is the outside-of-the-study area.

**Table 1 jof-11-00329-t001:** Diagnostic criteria of CPA by ERS/ESCMID.

Diagnostic Criteria	Features	Presence in Patients
Clinical criteria	Hemoptysis, cough, chest discomfort, dyspnea, or fatigue ≥ 3 months	Yes
Radiological criteria	Cavitation and/or the presence of a fungal ball	Yes
Microbiological criteria	Positive culture of *Aspergillus* spp. from respiratory specimens and/or positive *Aspergillus* spp. antibody (ELISA and/or ICT)	Yes

**Table 2 jof-11-00329-t002:** Demographics and clinical characteristics.

Characteristics	All (*n* = 219)	CPA (*n* = 144)	Non-CPA (*n* = 75)	*p*-Value
Gender ^a^				
Male	132 (60.3%)	82 (56.9%)	50 (66.7%)	
Female	87 (39.7%)	62 (43.1%)	25 (33.3%)	0.163
Age ^a^				
Mean (range)	53.63 (19–80)	54.07 (19–80)	52.77 (24–74)	0.185
Body mass index (BMI) [28], ^b^				
Median	18.9 (10.4–31.6)	18.4 (11.7–28.9)	19.7 (10.4–31.6)	0.135
Category ^a^				0.002
Underweight (<18.5 kg/m^2^)	100 (45.7%)	73 (50.7%)	27 (36%)	
Normoweight (18.5–22.9 kg/m^2^)	84 (38.4%)	46 (31.9%)	38 (50.7%)	
Overweight (23–24.9 kg/m^2^)	20 (9.1%)	18 (12.5%)	2 (2.7%)	
Obese (≥25 kg/m^2^)	15 (6.8%)	7 (4.9%)	8 (10.7%)	
Symptoms *^,a^				
Cough	194 (88.6%)	133 (92.4%)	61 (81.3%)	0.015
Fatigue	173 (78.9%)	115 (79.9%)	58 (77.3%)	0.663
Dyspnea	165 (75.3%)	110 (76.4%)	55 (73.3%)	0.619
Chest pain	143 (65.3%)	90 (62.5%)	53 (70.7%)	0.228
Hemoptysis	121 (55.3%)	83 (57.6%)	38 (50.7%)	0.325
Underlying diseases ^a^				
COPD	44 (20.1%)	30 (20.8%)	14 (18.7%)	0.704
Hypertension	28 (12.8%)	21 (14.6%)	7 (9.3%)	0.270
Asthma	18 (8.2%)	9 (6.3%)	9 (12%)	0.142
Diabetes mellitus	19 (8.7%)	12 (8.3%)	7 (9.3%)	0.803

* More than 3 months. ^a^: Chi-squared test was used for categorical variable comparisons. ^b^: Mann–Whitney test was applied for non-normally distributed continuous variables (BMI).

**Table 3 jof-11-00329-t003:** The positivity of fungal culture and serology across radiological patterns.

Radiological Findings	*n*	Positive Culture (%)			Positive Serology (%)	
		*A. fumigatus*	*A. flavus*	*A. niger*	ELISA	ICT
Infiltrate						
Positive	165	39 (23.6%)	17 (10.3%)	34 (20.6%)	62 (37.6%)	68 (41.2%)
Negative	54	15 (27.8%)	4 (7.4%)	10 (18.5%)	29 (53.7%)	32 (59.3%)
Cavitation						
Positive	191	49 (25.7%)	20 (10.5%)	42 (22%)	79 (41.4%)	87 (45.5%)
Negative	28	5 (17.9%)	1 (3.6%)	2 (7.1%)	12 (42.9%)	13 (46.4%)
Pericavitary fibrosis						
Positive	173	45 (26%)	18 (10.4%)	38 (22%)	76 (43.9%)	79 (45.7%)
Negative	46	9 (19.6%)	3 (6.5%)	6 (13%)	15 (32.6%)	21 (45.7%)
Pleural thickening						
Positive	95	26 (27.4%)	11 (11.6%)	24 (25.3%)	43 (45.3%)	42 (44.2%)
Negative	124	28 (22.6%)	10 (8.1%)	20 (16.1%)	48 (38.7%)	58 (46.8%)
Nodule						
Positive	30	7 (23.3%)	3 (10%)	4 (13.3%)	14 (46.7%)	13 (43.3%)
Negative	189	47 (24.9%)	18 (9.5%)	40 (21.2%)	77 (40.7%)	87 (46%)
Aspergilloma						
Positive	24	8 (33.3%)	1 (4.2%)	3 (12.5%)	16 (66.7%)	16 (66.7%)
Negative	195	46 (23.6%)	20 (10.3%)	41 (21%)	75 (38.5%)	84 (43.1%)
Bronchiectasis						
Positive	133	34 (25.6%)	17 (12.8%)	31 (23.3%)	62 (46.6%)	67 (50.4%)
Negative	86	20 (23.3%)	4 (4.7%)	13 (15.1%)	29 (33.7%)	33 (38.4%)

*n* = total number of patients; ICT = immunochromatographic tests.

**Table 4 jof-11-00329-t004:** CPA proportion by province.

	Province	CPA (*n* = 144)	Total Recruited (*n* = 219)	Proportion (%)
Region I	DKI Jakarta	30	42	71.4
	Banten	35	41	85.4
	West Java	21	32	65.6
Region II	West Sumatera	14	25	56
	Lampung	2	8	25
	Central Java	14	20	70
	East Java	21	40	52.5
	East Kalimantan	7	11	63.6

**Table 5 jof-11-00329-t005:** Distribution of *Aspergillus* species across regions.

Fungal Culture	Total (*n* = 219)	Region I (*n* = 115)	Region II (*n* = 104)	*p*-Value
*Aspergillus* spp.	81 (37%)	57 (49.6%)	24 (23.1%)	<0.001
*Aspergillus fumigatus*	54 (24.8%)	38 (33%)	16 (15.5%)	0.003
*Aspergillus flavus*	21 (9.6%)	14 (12.2%)	7 (6.8%)	0.179
*Aspergilllus niger*	44 (20.2%)	35 (30.4%)	9 (8.7%)	<0.001

## Data Availability

The original contributions presented in this study are included in the article. Further inquiries can be directed to the corresponding author.

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
