# Peer review of "A Silent Threat in Post-Tuberculosis Patients: Chronic Pulmonary Aspergillosis Survey in Multiple Regions of Indonesia (I-CHROME Study)"

_jof, 2025, doi:10.3390/jof11050329_

Round 1
Reviewer 1 Report
Dear authors,
Recently, a lot of effort has been invested in raising awareness and knowledge about the importance, possible diagnosis and treatment of invasive fungal infections, aspergillosis including. Despite this, strategy remains a major challenge, primarily in economically underdeveloped countries. You presented your work in a convincing way and obviously, much effort has been made in conducting the study. My opinion is that it can be published after minor revision.
Suggestions: General:
It will be necessary to explain how you were separate Aspergillus-infection from Aspergillus –colonization in cases of positive culture. Since immunodiagnostic had excellent negative predictive values, could you add these statistical parameters?
Specific: Sentence line 48-50: Serological tests included Aspergillus-specific IgG detection via ICT and ELISA, with sputum and blood samples analyzed for radiological and fungal findings-has to be improved it is not clear and not correct.
line 120: To provide a more in-depth picture of CPA burden in Indonesia, in post-tuberculosis patients, the study examines regional disparities in CPA proportion, microbiological profiles, and. -you have to finish sentence.
line 122 : you can improve these two sentences- This research contributes to the growing body of evidence supporting early and accurate diagnosis as a cornerstone of CPA care. These findings are essential for addressing the rising burden of CPA in Indonesia and other high-TB-burden countries.
Line 352- sentence - However, other symptoms, such as fatigue, dyspnoea, chest pain, and hemoptysis, were prevalent in both groups with no significant differences, indicating the importance of incorporating serological and mycological testing into the diagnostic algorithm for CPA - should be reformulated and emphasized well-known facts as the uncharacteristic symptoms of invasive fungal infections which influences that clinicians primarily considering possible bacteria as the causative agents.
Add a mark for the Chi-squared test below Tables 2 and 5.
Author Response
Reviewer 1:
Major Comments:
Recently, a lot of effort has been invested in raising awareness and knowledge about the importance, possible diagnosis and treatment of invasive fungal infections, aspergillosis including. Despite this, strategy remains a major challenge, primarily in economically underdeveloped countries. You presented your work in a convincing way and obviously, much effort has been made in conducting the study. My opinion is that it can be published after minor revision.
Suggestions: General:
It will be necessary to explain how you were separate Aspergillus-infection from Aspergillus –colonization in cases of positive culture. Since immunodiagnostic had excellent negative predictive values, could you add these statistical parameters?
Response:
We agree with the reviewer that distinguishing between Aspergillus infection and colonization is critical, especially in settings where Aspergillus spp. can be isolated from respiratory specimens without clear clinical disease. In our study, we adhered to the ERS/ESCMID diagnostic criteria for Chronic Pulmonary Aspergillosis (CPA), which require a combination of clinical symptoms ≥3 months, radiological abnormalities (such as cavitation, fibrosis, or aspergilloma), and microbiological evidence, including either a positive culture or serology.
To differentiate colonization from infection, patients with positive culture but without radiological features or chronic symptoms were not classified as CPA. Only those fulfilling all three diagnostic pillars—clinical, radiological, and microbiological/serological—were classified as CPA cases. This approach minimizes the misclassification of colonized individuals as infected.
In this paper, we served the data those from patient population specific for CPA. Therefore, the calculation of diagnostic performance for ELISA and ICT antibody were not shown.
However, if calculated, the NPV value as follows:
- For ICT, the NPV was 100%: none of the non-CPA patients had a positive ICT result.
- For ELISA, the NPV was also 100%, since no false positives were observed among the non-CPA cohort.
Detail comments:
- Specific: Sentence line 48-50: Serological tests included Aspergillus-specific IgG detection via ICT and ELISA, with sputum and blood samples analyzed for radiological and fungal findings-has to be improved it is not clear and not correct.
Response:
We appreciate the reviewer’s comment and have revised the sentence to improve clarity and accuracy. The original sentence was indeed imprecise in referring to the use of blood and sputum samples. The corrected sentences below:
“Serological tests for Aspergillus-specific IgG were conducted using both immunochromatographic test (ICT) and ELISA on serum samples. In parallel, sputum specimens were used for fungal culture, and radiological evaluations (e.g., chest X-rays or CT scans) were performed to identify typical CPA features such as cavitation and fibrosis.” (Line 47 – 50)
- Line 120: To provide a more in-depth picture of CPA burden in Indonesia, in post-tuberculosis patients, the study examines regional disparities in CPA proportion, microbiological profiles, and. -you have to finish sentence.
Response:
We have revised and completed it for clarity and coherence. The corrected version below:
“To provide a more in-depth picture of the CPA burden in Indonesia among post-tuberculosis patients, the study examines regional disparities in CPA proportion, microbiological profiles, and diagnostic characteristics across various provinces.” (Line 123-126)
- Line 122 : you can improve these two sentences- This research contributes to the growing body of evidence supporting early and accurate diagnosis as a cornerstone of CPA care. These findings are essential for addressing the rising burden of CPA in Indonesia and other high-TB-burden countries.
Response:
We have revised the sentences to improve their clarity, flow, and impact. The updated version below:
“This study reinforces the importance of early and accurate diagnosis in managing chronic pulmonary aspergillosis (CPA). The findings highlight the urgent need to improve CPA detection and treatment, especially in Indonesia and other countries with a high burden of tuberculosis.” (Line 126-129)
- Line 352- sentence - However, other symptoms, such as fatigue, dyspnoea, chest pain, and hemoptysis, were prevalent in both groups with no significant differences, indicating the importance of incorporating serological and mycological testing into the diagnostic algorithm for CPA - should be reformulated and emphasized well-known facts as the uncharacteristic symptoms of invasive fungal infections which influences that clinicians primarily considering possible bacteria as the causative agents.
Response:
We agree that the original sentence did not adequately highlight the clinical challenge posed by the non-specific presentation of CPA and other invasive fungal infections. We have reformulated the sentence to reflect this clinical reality:
“Other symptoms such as fatigue, dyspnea, chest pain, and hemoptysis, were common in both CPA and non-CPA groups, showing no significant differences. These non-specific manifestations are typical of various pulmonary conditions, often leading clinicians to suspect bacterial etiologies initially. This diagnostic ambiguity underscores the necessity of incorporating specific serological and mycological tests into the diagnostic workup to improve early detection of CPA and avoid misdiagnosis.” (Line 383-389)
- Add a mark for the Chi-squared test below Tables 2 and 5.
Response:
We have added a note below both Table 2 and Table 5.
Reviewer 2 Report
The manuscript titled “Chronic Pulmonary Aspergillosis Survey in Multiple Regions in Indonesia: A Silent Threat in Post Tuberculosis Patients (I-CHROME Study)” provides a comprehensive and informative analysis of key epidemiological, clinical, and microbiological characteristics of CPA by analyzing post-TB patients in a high-burden setting. The authors have effectively communicated their findings, making a valuable contribution to the field. However, there are a few points that need further consideration and clarification:
Major points
- Materials and Methods, line 133: Please specify the exclusion criteria used in the study to ensure clarity and avoid any confusion.
- Table 2: The p-value for cough (p = 0.015) appears to be incorrect. If the chi-square test was applied, it should be approximately 0.4, which is not significant. Please verify this and check all p-values for symptoms, to ensure accuracy. The text needs to be revised accordingly.
- Table 2: The manuscript mentions underlying diseases for only 109 patients. What underlying conditions did the remaining 110 patients have? Please clarify.
- Results, lines 302-304: Could you explain the rationale behind dividing patients into Region I and Region II? The classification appears arbitrary—on what criteria was it based? Why was a direct comparison of different regions not sufficient?
- Figures 1 and 3, as well as all tables, are not mentioned in the text. Additionally, Figure 2 is incorrectly referenced as Figure 3. Please revise accordingly.
- Table 2: Please include the age range.
- Table 3: It would be helpful to add p-value columns next to IgG and ICT results for better readability and data interpretation.
- Discussion, lines 363-364. You state that your study strongly supports the diagnostic accuracy of the Aspergillus IgG test, with high positivity in CPA patients using ICT and ELISA, while all non-CPA patients tested negative. However, the specificity, sensitivity, and accuracy of the method are not presented in the Results section. Please consider including these data for completeness.
- One of the critical diagnostic markers of CPA was radiological findings, including cavitation and pericavitary fibrosis. Given the significant associations found between infiltrates and IgG (p = 0.037) or ICT (p = 0.021), these results merit inclusion in the Abstract to highlight their importance.
- Lines 365-366: You mention that 15 patients passed away, but Figure 1 states that one patient was excluded due to death. Was death an exclusion criterion? Please clarify this inconsistency.
Minor points
- Abstract, line 53: The abbreviation ICT should also be written out in full.
- Lines 48-50: The sentence is unclear. Do you mean “…with sputum and blood samples analyzed for associations with radiological and fungal findings”? Please consider rewording for clarity.
- Lines 50-51: This section should be revised to reflect the updated data accordingly.
- Introduction, lines 120-122: The sentence appears incomplete and should be revised for clarity.
- Lines 150-151. A reference would be helpful.
- Line 191: Do you mean "negative TB status"? Please clarify.
- Figure – Aspergillus The description of the image should be improved. Consider specifying the culture medium (e.g., colonies on Sabouraud dextrose agar) and adding arrows to indicate the relevant colonies, making it easier for readers unfamiliar with culture morphology to interpret.
- Line 370: The phrase "perfect sensitivity" may not be the most appropriate wording. Consider rephrasing for accuracy (e.g. high sensitivity, or x% sensitivity).
- Line 395: The percentage 32.6% appears to be incorrect. Based on Table 3, it should be 42.9%. Please verify and correct this discrepancy.
- Lines 413-414: The sentence is unclear. Please rephrase for clarity.
- The Abstract should be revised to reflect all modifications.
- Under Data Availability Statement, lines 443-448: The actual statement is missing, and only the MDPI instructions are shown. Please include the required information.
- Lines 427-428: The sentence “For research articles with several authors, … statements should be used” can be removed, as it is part of the journal’s instructions.
Author Response
Reviewer 2:
Major Comments:
The manuscript titled “Chronic Pulmonary Aspergillosis Survey in Multiple Regions in Indonesia: A Silent Threat in Post Tuberculosis Patients (I-CHROME Study)” provides a comprehensive and informative analysis of key epidemiological, clinical, and microbiological characteristics of CPA by analyzing post-TB patients in a high-burden setting. The authors have effectively communicated their findings, making a valuable contribution to the field. However, there are a few points that need further consideration and clarification:
Major points:
- Materials and Methods, line 133: Please specify the exclusion criteria used in the study to ensure clarity and avoid any confusion.
Response:
We have revised the Materials and Methods section to clearly define the exclusion criteria applied in the study. The updated text now reads:
“Exclusion criteria included: (1) refusal to provide informed consent for participation or serological testing, (2) incomplete clinical or diagnostic data, (3) current diagnosis or ongoing treatment for active tuberculosis at the time of screening, (4) death prior to or during diagnostic evaluation, (5) pregnancy, (6) known HIV infection, (7) severe immunocompromised state (e.g., recent chemotherapy with complications or critical illness), and (8) use of systemic antifungal agents within the last one month.”
This clarification has been inserted in the “Study Design and Patient Selection” subsection (Line 137-142).
- Table 2: The p-value for cough (p= 0.015) appears to be incorrect. If the chi-square test was applied, it should be approximately 0.4, which is not significant. Please verify this and check all p-values for symptoms, to ensure accuracy. The text needs to be revised accordingly.
Response:
All corrected data sets and p-values have been updated in Table 2 and annotated in the revised manuscript. We have also ensured that all statistical tests used are indicated.
“The analysis of symptoms revealed that cough was significantly more common in CPA patients, with 92.4% reporting this symptom compared to 81.3% in the non-CPA group (p = 0.015).” (Line 222-224).
- Table 2: The manuscript mentions underlying diseases for only 109 patients. What underlying conditions did the remaining 110 patients have? Please clarify.
Response:
The apparent discrepancy is due to the fact that only 109 of the 219 patients had at least one recorded underlying disease, such as COPD, hypertension, asthma, or diabetes mellitus. The remaining 110 patients did not have any documented underlying comorbidities, based on medical history and clinical evaluation at the time of enrollment. To avoid confusion, we have revised the corresponding section in the manuscript to clarify this:
“Regarding underlying diseases, 109 (49.8%) had at least one underlying condition, including COPD, hypertension, asthma, or diabetes mellitus. The remaining 110 (50.2%) patients had no documented underlying comorbidities.” (Line 238-240)
- Results, lines 302-304: Could you explain the rationale behind dividing patients into Region I and Region II? The classification appears arbitrary—on what criteria was it based? Why was a direct comparison of different regions not sufficient?
Response:
The division into Region I (Jakarta, Tangerang, and West Java) and Region II (Central Java, East Java, Sumatra, and Kalimantan) was based on geographic proximity, socioeconomic characteristics, and differences in healthcare accessibility. Region I includes more urbanized and densely populated areas with advanced healthcare infrastructure and higher diagnostic capacity. Region II, in contrast, consists of more rural or less densely populated areas with relatively limited access to specialized diagnostic tools. (Line 324-330)
- Figures 1 and 3, as well as all tables, are not mentioned in the text. Additionally, Figure 2 is incorrectly referenced as Figure 3. Please revise accordingly.
Response:
All figures (Figures 1–3) and tables (Tables 1–5) are now explicitly cited in the main text at appropriate locations to ensure proper narrative and data presentation alignment.
- Table 2: Please include the age range.
Response:
We have updated Table 2 to include the age range for both CPA and non-CPA groups, as well as for the overall study population. The revised table now includes the following row under the “Age” category:
Range: 19–80 years (CPA: 19–80; Non-CPA: 24–74)
- Table 3: It would be helpful to add p-value columns next to IgG and ICT results for better readability and data interpretation.
Response:
We acknowledge the value of adding p-value columns for statistical comparison. However, in this study, the data presented are derived from a population of patients who already exhibited clinical and radiological features strongly suggestive of CPA. As such, the focus was on describing the distribution of serological results across radiological findings rather than evaluating diagnostic test performance in a general or screening population. Therefore, we chose not to calculate or include p-values for each IgG and ICT result in Table 3, as these could be misinterpreted outside the context of the pre-selected diagnostic cohort. This approach aligns with the study’s objective to describe characteristics within a clinically enriched CPA population rather than perform a comparative diagnostic evaluation.
- Discussion, lines 363-364. You state that your study strongly supports the diagnostic accuracy of the AspergillusIgG test, with high positivity in CPA patients using ICT and ELISA, while all non-CPA patients tested negative. However, the specificity, sensitivity, and accuracy of the method are not presented in the Results section. Please consider including these data for completeness.
Response:
This study was conducted on a population of patients who already presented with clinical and radiological features highly suggestive of CPA. Therefore, the study population is not representative of a general or undifferentiated clinical setting, but rather an enriched cohort already selected based on pre-test suspicion.
For this reason, we chose not to present formal diagnostic performance calculations (sensitivity, specificity, PPV, NPV) for ELISA and ICT in the Results section, as these values might be misleading when interpreted outside the context of a prospective screening or diagnostic accuracy study. Instead, we focused on describing the positivity rates and clinical value of serological tests in supporting CPA diagnosis among clinically preselected cases.
- One of the critical diagnostic markers of CPA was radiological findings, including cavitation and pericavitary fibrosis. Given the significant associations found between infiltrates and IgG (p = 0.037) or ICT (p = 0.021), these results merit inclusion in the Abstract to highlight their importance.
Response:
We agree that the significant associations between radiological infiltrates and positive serological results (IgG and ICT) are important diagnostic findings. To reflect their relevance, we have updated the Abstract with the following sentence:
“Among those with infiltrates, a positive Aspergillus culture was not more common (p>0.05) whereas Aspergillus IgG was more often raised (p=0.037) as was a positive ICT (p=0.021).” (Line 55-57)
- Lines 365-366: You mention that 15 patients passed away, but Figure 1 states that one patient was excluded due to death. Was death an exclusion criterion? Please clarify this inconsistency.
Response:
We clarify that one patient was excluded at baseline due to death before diagnostic evaluation. An additional 15 CPA patients died during longitudinal follow-up; these outcomes will be reported in a subsequent study.
Detail Comments:
Minor points
- Abstract, line 53: The abbreviation ICT should also be written out in full.
Response:
Thank you for this suggestion. We have revised the Abstract to spell out the abbreviation upon first mention. The updated sentence now reads:
“Aspergillus-specific IgG testing demonstrated high diagnostic utility, with positivity rates of 69.4% for immunochromatographic test (ICT) and 63.2% for ELISA among CPA patients.” (Line 54-55)
- Lines 48-50: The sentence is unclear. Do you mean “…with sputum and blood samples analyzed for associations with radiological and fungal findings”? Please consider rewording for clarity.
Response:
Yes, your interpretation is correct, and we agree that the original sentence lacked clarity.
We have revised it as follows:
“Sputum specimens were used in parallel for fungal culture, and radiological evaluations (e.g., chest X-rays or CT scans) were performed to identify typical CPA features such as cavitation and fibrosis.”(Line 48-50)
- Lines 50-51: This section should be revised to reflect the updated data accordingly.
Response:
We have revised this part of the text to align with the corrected and updated results presented in the manuscript. The updated sentence is below:
“Aspergillus-specific IgG testing demonstrated high diagnostic utility, with positivity rates of 69.4% for ICT and 63.2% for ELISA among CPA patients. Among those with infiltrates, a positive Aspergillus culture was not more common (p>0.05), whereas Aspergillus IgG was more often raised (p=0.037) as was a positive ICT (p=0.021).” (Line 54-57)
- Introduction, lines 120-122: The sentence appears incomplete and should be revised for clarity.
Response:
We have revised the sentence for completeness and clarity. The corrected version is below:
“To provide a more in-depth picture of the CPA burden in Indonesia among post-tuberculosis patients, the study examines regional disparities in CPA proportion, microbiological profiles, and diagnostic characteristics across various provinces.” (Line 123-126)
- Lines 150-151. A reference would be helpful.
Response:
We have added an appropriate reference (Line 169 ) as follows:
Denning DW, Cadranel J, Beigelman-Aubry C, et al. Chronic pulmonary aspergillosis: rationale and clinical guidelines for diagnosis and management. Eur Respir J 2016; 47(1):45-68.
- Line 191: Do you mean "negative TB status"? Please clarify.
Response:
Yes, the intended meaning was “negative TB status.” We have revised the sentence for clarity. The updated version now reads:
“A total of 221 patients aged 18 years or older, with a prior history of tuberculosis (TB), patients with negative TB, and respiratory symptoms lasting at least three months were screened for inclusion in the study.” (Line 197-199)
- Figure – AspergillusThe description of the image should be improved. Consider specifying the culture medium (e.g., colonies on Sabouraud dextrose agar) and adding arrows to indicate the relevant colonies, making it easier for readers unfamiliar with culture morphology to interpret.
Response:
We have revised the figure legend to include the culture medium used and added arrows in the image to clearly indicate the Aspergillus fumigatus and Aspergillus niger colonies. The updated figure caption is below:
Figure 2. A culture plate of Sabouraud dextrose agar from a CPA patient. The dark brown colonies were consistent with the Aspergillus niger complex. The grey green colony shows white areas represent poorly sporulating sections within the colony (characteristic of many CPA isolates) and microscopic examination revealed Aspergillus fumigatus complex.
(Line 277-280)
- Line 370: The phrase "perfect sensitivity" may not be the most appropriate wording. Consider rephrasing for accuracy (e.g. high sensitivity, or x% sensitivity).
Response:
We have revised:
If CPA was diagnosed only with combination of clinical, radiology and sputum culture, 70 (61.4%) cases were undetectable as CPA cases. These findings showed proof that serology test such as ICT and/or ELISA could serve as an effective diagnostic method in endemic regions, especially ICT for rural areas with limited laboratory infrastructure. (Line 401-405).
- Line 395: The percentage 32.6% appears to be incorrect. Based on Table 3, it should be 42.9%. Please verify and correct this discrepancy.
Response:
Upon verification with Table 3, you are absolutely correct—the percentage of IgG positivity among patients without cavitation is 42.9%, not 32.6% as previously stated. We have corrected the value in the Discussion section (Line 395) to maintain consistency and accuracy:
“Among patients without cavitations, serological positivity for IgG and ICT was 42.9% and 46.4%, respectively.” (Line 436-437)
- Lines 413-414: The sentence is unclear. Please rephrase for clarity.
Response:
Thank you for your comment. Upon review, we found that the content of the sentence in Lines 413–414 had already been adequately addressed in the preceding paragraph and summarized in the conclusion section. To avoid redundancy and improve the flow of the discussion section, we have removed the sentences from the final version of the manuscript.
- The Abstract should be revised to reflect all modifications.
Response:
We have revised the Abstract (Line 40-62) as follows:
Abstract: Background: A significant complication among post-tuberculosis patients is chronic pulmonary aspergillosis (CPA), with prevalence and outcomes varying by region. This study aimed to explore the epidemiology, clinical characteristics, and microbiological profiles of 219 post-tuberculosis patients with persistent respiratory symptoms and lung cavities in Indonesia. Methods: The patients were divided into CPA (n=144) and non-CPA (n=75) groups. This cross-sectional study diagnosed CPA in post-tuberculosis patients using ERS/ESCMID criteria, integrating clinical, radiological, and fungal assessments. Serological tests for Aspergillus-specific IgG were conducted using immunochromatographic (ICT) and ELISA on serum samples. In parallel, sputum specimens were used for fungal culture, and radiological evaluations (e.g., chest X-rays or CT scans) were performed to identify typical CPA features such as cavitation and fibrosis. Results: Persistent cough was significantly more common in CPA patients (83.3%, p=0.015), highlighting its role as a clinical indicator for CPA. Radiological infiltrates were found in 165 patients (75.3%); critical diagnostic markers of CPA were cavitation and pericavitary fibrosis. Aspergillus-specific IgG testing demonstrated high diagnostic utility, with positivity rates of 69.4% for ICT and 63.2% for ELISA among CPA patients. Among those with infiltrates, a positive Aspergillus culture was not more common (p>0.05) whereas Aspergillus IgG was more often raised (p=0.037) as was a positive ICT (p=0.021). Regional analysis revealed a higher CPA burden in Region 1 (75%) compared to Region 2 (56%, p=0.003), with Aspergillus fumigatus and Aspergillus niger predominating in Region 1. Conclusion: These findings highlight the importance of comprehensive approaches and region-specific CPA management strategies in Indonesia.
Keywords: chronic pulmonary aspergillosis; post-tuberculosis; IgG antibodies
- Under Data Availability Statement, lines 443-448: The actual statement is missing, and only the MDPI instructions are shown. Please include the required information.
Response:
Thank you for highlighting this oversight. We have replaced the placeholder text with the appropriate Data Availability Statement. The revised section now reads:
“Data supporting the findings of this study are available from the corresponding author upon reasonable request.” (Line 477-478)
- Lines 427-428: The sentence “For research articles with several authors, … statements should be used” can be removed, as it is part of the journal’s instructions.
Response:
We have removed the instructional placeholder text from the Author Contributions section to comply with the journal's formatting requirements. (Line 463-469).